# Posttraumatic Cutaneous Meningioma with a “Meningiolipoma” Pattern Presenting as a Nasal Bridge Mass

**DOI:** 10.3390/diagnostics14161731

**Published:** 2024-08-09

**Authors:** Dong Ren, Jerry Lou, Edward Kuan, Mari Perez-Rosendahl, William H. Yong

**Affiliations:** 1Department of Pathology and Laboratory Medicine, University of California, Irvine, CA 92697, USA; dren3@hs.uci.edu (D.R.); jjlou@hs.uci.edu (J.L.); mperezro@hs.uci.edu (M.P.-R.); 2Department of Otolaryngology-Head and Neck Surgery, University of California, Irvine, CA 92697, USA; eckuan@hs.uci.edu; 3Department of Pathology, University of California, Irvine Medical Center, Orange, CA 92868, USA

**Keywords:** extracranial cutaneous meningioma, nasal bridge, head trauma

## Abstract

Meningiomas are tumors originating from arachnoid meningothelial cells. Occasionally, meningiomas are identified outside the central nervous system, and are referred to as extracranial meningiomas (EMs). The vast majority of EMs are an extension from an intracranial or intraspinal tumor. However, primary EMs may arise from extracranial sites with the most common sites being the skin and scalp subcutis, which are further categorized as cutaneous meningiomas (CMs). CMs are rare cutaneous tumors with similar ultrastructural and cytologic findings compared to those of intracranial meningiomas, but with a wide range of histologic differences. Therefore, an assessment using a panel of investigative tools, including imaging, histopathology, and immunohistochemistry, is required to determine the diagnosis of CMs. Here, we report the case of a 64-year-old gentleman presenting with a posttraumatic well-circumscribed superficial mass overlying the right nasal bridge. We are unable to identify other cases arising in the nasal bridge.

Meningiomas are the most frequently reported primary central nervous system (CNS) neoplasms in adults, accounting for over one-third of all CNS tumors [1]. They are classified as intracranial extra-axial neoplasms by the World Health Organization (WHO) due to their origin from arachnoid meningothelial cells. While most intracranial meningiomas are typically diagnosed via medical imaging and subsequent histopathologic examination, extracranial meningiomas are easily misdiagnosed due to their rarity and a broad spectrum of differential diagnoses with benign nerve sheath tumors and skin tumors. The diagnosis is typically made only after histopathologic examination in most cases [2,3]. Complete surgical excision is the first-line treatment for the majority of symptomatic and enlarging extracranial meningiomas, and patients exhibit favorable prognosis and a low recurrence rate [4].

Extracranial meningiomas can be divided into primary and secondary extracranial meningiomas (PEMs and SEMs) based on the tumor origin, where SEMs present at a secondary location away from an existing primary intracranial tumor, whereas PEMs refer to the meningiomas that arise from extracranial sites without a known primary intracranial tumor [5]. The common sites for PEMs are the head and neck region, including the skin and the scalp subcutis; sinonasal PEMs are less common [6]. Although extracranial meningiomas are rare, extension and metastasis from intracranial or intraspinal tumors comprise the vast majority of extracranial meningiomas. Therefore, the exclusion of a primary intracranial meningioma is required for the diagnosis of a PEM.

Primary cutaneous meningiomas are usually present at birth, with a high occurrence on the scalp, forehead, and paravertebral regions [7]; these are designated as type I lesions thought to derive from ectopic arachnoid cells [8]. Type II lesions usually present in adulthood and are derived from meningothelial cells located in the cutis. Type II lesions have been speculated to arise from ectopic meningothelial cells, potentially displaced from the arachnid during head trauma. Type III lesions are defined by direct extension from intracranial or intraspinal tumors [8]. Herein, we report an unusual case of a Type II cutaneous meningioma involving the nasal bridge that may be associated with remote head trauma (Figure 1 and Figure 2).

## Figures and Tables

**Figure 1 diagnostics-14-01731-f001:**
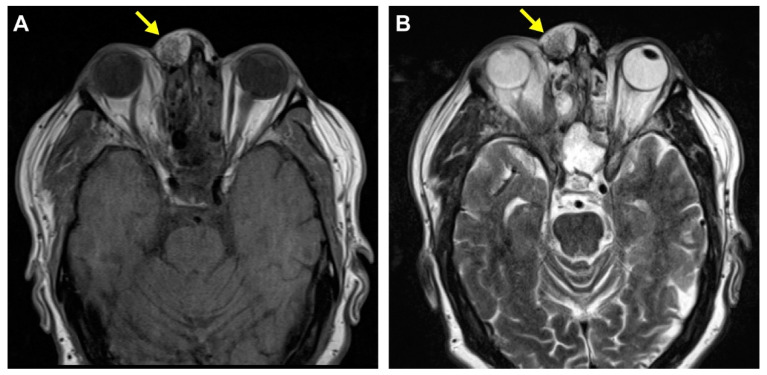
Radiologic finding of the mass on MRI imaging. A 64-year-old man presented with a slightly tender mass overlying the right nasal bridge following back surgery three years ago. An axial T1 non-contrast CT scan revealed a well-circumscribed soft tissue mass external to the right nasal bone with no intracranial or sinus communication (**A**). An MRI scan showed a recently enlarging 2.0 × 1.4 × 1.1 cm circumscribed superficial mass with heterogeneous enhancement and mixed fat and soft tissue signal intensity components overlying the right nasal bridge (**B**).

**Figure 2 diagnostics-14-01731-f002:**
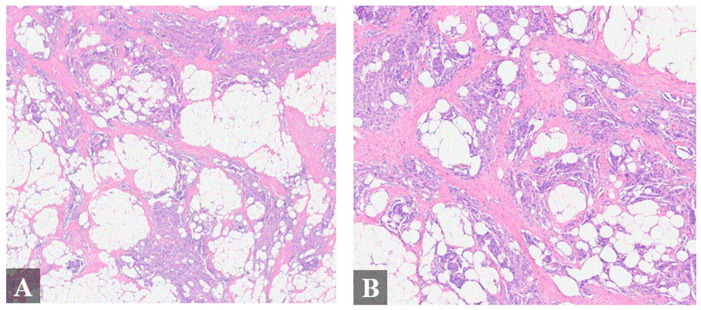
Histologic finding of the resected mass. Histologic examination of the specimen showed a meningothelial vascular-rich epithelial proliferation arranged in nodular nests heterogeneously distributed within fibroadipose tissue and skeletal muscle (**A**–**D**), resembling a glomus tumor or “meningiolipoma”. (**A**), 4× magnification; (**B**), 4× magnification; (**C**), 20× magnification; (**D**), 20× magnification. Immunohistochemistry (IHC) exhibited immunoreactivity for epithelial membrane antigen (EMA), progesterone receptor ((**E**), 20× magnification), somatostatin receptor 2 ((**F**), 20× magnification), and androgen receptor. Cytokeratin AE1/AE3, p63, S100, smooth muscle actin, desmin, and HMB-45 were negative.

## Data Availability

The authors declare that all the data described in this article are available upon reasonable request.

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
