# Peer review of "Posttraumatic Cutaneous Meningioma with a “Meningiolipoma” Pattern Presenting as a Nasal Bridge Mass"

_diagnostics, 2024, doi:10.3390/diagnostics14161731_

Round 1

Reviewer 1 Report

Comments and Suggestions for Authors

Ren et al briefly present an “interesting images case” of an extracranial meningioma in the form of a primary cutaneous meningioma classified as a type II cutaneous meningioma involving the nasal bridge. T1 noncontrast and T2 images were shown as was pathology. This demonstrates radiographic characteristics of a rare tumor. The only change I would recommend in wording is on lines 23-24. Technically, meningiomas are not diseases of the central nervous system as they arise from dura. They are intracranial tumors and I would word the first sentence in the introduction as "meningiomas are the most frequently reported intracranial neoplasm in adults, accounting for over one third of all intracranial tumors." Otherwise, this is a brief and well written description of a rare entity demonstrating its radiographic and pathology characteristics. Only requires minor changes as detailed above. 

Author Response

Reviewer #1 (Comments to the Author (Required)): 
Ren et al briefly present an “interesting images case” of an extracranial meningioma in the form of a primary cutaneous meningioma classified as a type II cutaneous meningioma involving the nasal bridge. T1 noncontrast and T2 images were shown as was pathology. This demonstrates radiographic characteristics of a rare tumor. The only change I would recommend in wording is on lines 23-24. Technically, meningiomas are not diseases of the central nervous system as they arise from dura. They are intracranial tumors and I would word the first sentence in the introduction as "meningiomas are the most frequently reported intracranial neoplasm in adults, accounting for over one third of all intracranial tumors." Otherwise, this is a brief and well written description of a rare entity demonstrating its radiographic and pathology characteristics. Only requires minor changes as detailed above.
Responses: We do appreciate the reviewer for raising this important point and the reviewer’s point is really valuable and helpful. As requested, appropriate correction has been made and highlighted in Red font in the “Introduction” section of the revised manuscript.

Reviewer 2 Report

Comments and Suggestions for Authors

The present paper presents a cutaneous meningioma identified as a nasal bridge mass in MRI investigations. The authors provide sufficient information in the Introduction section related to cutaneous meningiomas, which have distinct features and classifications, making the presented case to be included in Type II cutaneous meningioma.

The MRI images are sharp and the localization of the tumor is very clear seen in Figure 1.

Some modification should be made to Figure 2:

-        Please provide a better quality of Figure 2 (A and B), as the contrast between hematoxylin and eosin staining is dim. 

-        Could you please add pictures of the negative staining; being a paper about Interesting images, I suggest adding to Figure 2 two more pictures of the negative markers (of your choice – cytokeratin, p63, alpha-SMA, desmin, HMB-45), which will considerably improve the quality of the paper.  

Author Response

Reviewer #2 (Comments to the Author (Required)): 
The present paper presents a cutaneous meningioma identified as a nasal bridge mass in MRI investigations. The authors provide sufficient information in the Introduction section related to cutaneous meningiomas, which have distinct features and classifications, making the presented case to be included in Type II cutaneous meningioma.

The MRI images are sharp and the localization of the tumor is very clear seen in Figure 1.

Some modification should be made to Figure 2:
-        Please provide a better quality of Figure 2 (A and B), as the contrast between hematoxylin and eosin staining is dim. 
-        Could you please add pictures of the negative staining; being a paper about Interesting images, I suggest adding to Figure 2 two more pictures of the negative markers (of your choice – cytokeratin, p63, alpha-SMA, desmin, HMB-45), which will considerably improve the quality of the paper.
Responses: We do appreciate the reviewer for raising these important points and the reviewer’s points are well taken As requested by the reviewer, we re-took the images with better quality and resolution for Figure 2A and B. In terms of the negative staining, we did not find the slides in the file storage since our department is remodeling now and the slides from last year was stored at our department at this point. Unfortunately, we can not take pictures for the negative staining at this point. Sorry for any inconvenience this may have caused.